# Glucose Concentration in Cell Culture Medium Influences the BRCA1-Mediated Regulation of the Lipogenic Action of IGF-I in Breast Cancer Cells

**DOI:** 10.3390/ijms21228674

**Published:** 2020-11-17

**Authors:** Moses O. Koobotse, Dayane Schmidt, Jeff M. P. Holly, Claire M. Perks

**Affiliations:** 1IGFs & Metabolic Endocrinology Group, Bristol Medical School, Translational Health Sciences, University of Bristol, Bristol BS10 5NB, UK; koobotsem@ub.ac.bw (M.O.K.); dayschmidt5@gmail.com (D.S.); jeff.holly@bristol.ac.uk (J.M.P.H.); 2Faculty of Health Sciences, School of Allied Health Professions, University of Botswana, Gaborone, Plot 4775, Botswana

**Keywords:** hyperglycaemia, BRCA1, IGF-I, fatty acid synthesis, breast cancer

## Abstract

Hyperglycaemia is a common metabolic alteration associated with breast cancer risk and progression. We have previously reported that BRCA1 restrains metabolic activity and proliferative response to IGF-I anabolic actions in breast cancer cells cultured in high glucose. Here, we evaluated the impact of normal physiological glucose on these tumour suppressive roles of BRCA1. Human breast cancer cells cultured in normal physiological and high glucose were treated with IGF-I (0–500 ng/mL). Cellular responses were evaluated using immunoblotting, co-immunoprecipitation, and cell viability assay. As we previously reported, IGF-I induced ACCA dephosphorylation by reducing the association between BRCA1 and phosphorylated ACCA in high glucose, and upregulated FASN abundance downstream of ACCA. However, these effects were not observed in normal glucose. Normal physiological glucose conditions completely blocked IGF-I-induced ACCA dephosphorylation and FASN upregulation. Co-immunoprecipitation studies showed that normal physiological glucose blocked ACCA dephosphorylation by increasing the association between BRCA1 and phosphorylated ACCA. Compared to high glucose, the proliferative response of breast cancer cells to IGF-I was reduced in normal glucose, whereas no difference was observed in normal mammary epithelial cells. Considering these results collectively, we conclude that normal physiological glucose promotes the novel function of BRCA1 as a metabolic restraint of IGF-I actions. These data suggest that maintaining normal glucose levels may improve BRCA1 function in breast cancer and slow down cancer progression.

## 1. Introduction

Breast cancer is the most frequently diagnosed cancer among women, accounting for approximately a quarter of all cancers and is also the most common cause of cancer death among women [1]. The majority of breast cancers are sporadic while 5–10% are familial, and approximately 40–50% of the familial breast tumours are associated with germ-line mutations in *breast cancer 1 early onset (BRCA1)* gene [2,3,4]. While *BRCA1* mutations are rare in sporadic breast cancers, BRCA1 dysfunction has also been reported in 24–63% of these cases and is associated with reduced or complete loss of BRCA1 mRNA and protein abundance [5,6,7], BRCA1 protein mis-localisation [2,8], as well as *BRCA1* gene promoter methylation [9].

Epidemiological evidence has indirectly linked BRCA-associated breast cancer risk with energy metabolic pathways. Risk estimates in Ashkenazi Jewish breast cancer patients with *BRCA1/2* mutations suggested that engaging in physical exercise and maintaining healthy body weight at adolescence significantly delayed BRCA1/2-associated breast cancer onset [3]. In addition, the risk of breast cancer in probands carrying *BRCA1/2* gene mutations has been changing over the years and this has been attributed to modifiable lifestyle-related factors [3,10,11]. In women who carry *BRCA1/2* gene mutations, high energy intake is associated with increased breast cancer risk compared to women with low energy intake [12]. BRCA1 is a tumour suppressor with several genomic functions that maintain genome integrity, including DNA repair and chromatin remodelling [13,14]. Therefore, data from the aforementioned epidemiological studies cannot be fully explained by these well-known functions of BRCA1 in maintaining genomic integrity. Beyond the genome however, BRCA1 has recently been linked to regulation of metabolic functions, particularly as a negative regulator of the fatty acid synthesis pathway.

At a cellular level, BRCA1 inhibits endogenous fatty acid synthesis by binding to the inactive form of acetyl CoA carboxylase (ACCA), a key enzyme of the fatty acid synthesis pathway [15]. Consistent with its role as a negative regulator of fatty acid synthesis, our previous study showed that BRCA1 inhibited lipogenic actions of insulin-like growth factor I (IGF-I) in breast cancer cells [16]. IGF-I is a member of the IGF axis, which plays a role in cell metabolism and is deregulated in metabolic disorders and cancer, including breast tumours [17,18]. Therefore, our data showing that BRCA1 inhibits IGF-I actions supports the role of BRCA1 in the regulation of metabolism. In other studies, downregulation of BRCA1 in myotubes increased intracellular lipid content [19] and transfection of *BRCA1*-mutant SUM1315 cell line with wildtype *BRCA1* resulted in downregulation of ATP citrate lyase (ACL) and reduction of free fatty acids [20]. In in vivo studies, BRCA1 was upregulated in adipose tissue from obese subjects together with phosphorylated ACCA and this data is consistent with the role of BRCA1 in limiting fatty acid synthesis during obesity [21]. In the absence of any breast abnormality, *BRCA1* germline mutation carriers exhibited alterations in lipid profiles in breast tissues, including a 19% increase in triglycerides and unsaturated lipids [22].

Breast cancer, obesity, and type 2 diabetes share common metabolic alterations including hyperglycaemia, which is associated with poor patient outcomes in cancer [23,24,25]. Hyperglycaemia, defined as excess serum glucose (>7 mM) [26], may develop from insulin resistance in both obesity and type 2 diabetes [17,27,28]. Since increased glucose uptake and utilisation are some of the hallmarks of cancer [29], hyperglycaemia may support cancer cells with an abundant supply of glucose [27]. Excess glucose is converted into fatty acids in the liver and cancer cells have been shown to activate the fatty acid synthesis pathway by upregulating lipogenic enzymes such as fatty acid synthase (FASN), ACCA, and ACL [30,31]. Hyperglycaemia may also influence cancer cells indirectly by increasing circulating IGF-I levels, secondary to chronic hyperinsulinemia [17,25,27]. Higher circulating levels of IGF-I have been associated with increased risk of several cancers including breast cancer [17].

Glucose is a soluble sugar added to cell culture media in concentrations ranging from 1 g/L (5.5 mM) to as high as 10 g/L (55 mM). Supplementation with approximately 5.5 mM d-glucose approximates normal blood sugar levels, while concentrations of glucose around 10 mM and above 10 mM are analogous to pre-diabetic and diabetic levels respectively [24,32]. Although high glucose concentrations are known to have detrimental effects on many cell types, it is not uncommon for media to contain diabetic levels of glucose supplementation, nor scientists to study the effect of such glucose levels [33,34,35]. Our previous study [16] was performed in Dulbecco’s modified Eagle’s medium (DMEM) supplemented with 25 mM of d-glucose, which is conventional for MCF7 and T47D cells. In that study, we showed that IGF-I reduces BRCA1 function in lipogenesis by reducing the interaction between BRCA1 and phosphorylated ACCA. In the present study, we investigated the influence of glucose levels by comparing data generated when using DMEM supplemented with either 5 mM (later referred as normal) or 25 mM (later referred as high) glucose.

We confirmed our previous findings in high glucose and concluded that IGF-I impairs BRCA1 function by reducing its interaction with phosphorylated ACCA, leading to ACCA dephosphorylation, FASN upregulation, and cell proliferation. These findings were not observed with normal levels of glucose in the current study. Our new findings show that unlike high glucose, normal glucose conditions promote BRCA1 function to regulate cell proliferation by restraining lipogenic actions of IGF-I. The data suggest that maintaining normal glucose levels supports BRCA1 function in restraining IGF-I lipogenic actions and that this may slow breast cancer progression.

## 2. Results

### 2.1. Glucose Level Regulates Key Proteins Involved in Endogenous Fatty Acid Synthesis in MCF7 Breast Cancer Cells

We have previously shown in estrogen receptor (ER)-positive breast cancer cell lines that IGF-I promoted lipogenesis by reducing the association between BRCA1 and the inactive, phosphorylated form of ACCA, resulting in de-phosphorylation of the enzyme [16]. Although these experiments were conducted under typical cell culture conditions, the glucose level in culture media is usually higher than normal physiological glucose (5–7 mM), typically at 25 mM, which is in the hyperglycaemic range (>7 mM) and corresponds to severe diabetes [26,33,36]. To investigate the impact of glucose levels on fatty acid synthesis, we cultured cells in normal glucose growth media supplemented with foetal bovine serum (FBS) for 24 h after which cells were switched to either normal or high glucose serum-free media for 24 and 48 h before analysis.

Compared to normal glucose, the abundances of phosphorylated ACCA and total ACCA were higher in high glucose in MCF7 cells after 24 and 48 h (Figure 1A–C). However, when phosphorylated ACCA was normalised to total ACCA, there was no difference in normalised ACCA phosphorylation between normal and high glucose conditions. Downstream of ACCA, the abundance of FASN increased 1.3-fold after 24 h (*p* < 0.05) (Figure 1D,E) and 1.4-fold after 48 h (Appendix A). AMP-activated protein kinase (AMPK) is a well-established ACCA kinase, therefore we expected that AMPK phosphorylation will parallel ACCA phosphorylation, however, this was not observed in our model (Figure 1D,F) (Appendix A). Since BRCA1 has been shown to bind phosphorylated ACCA, we considered the possibility that BRCA1 inhibited ACCA dephosphorylation in high glucose, resulting in an increase in the phosphorylated form of ACCA. Compared to normal glucose, BRCA1 abundance increased after exposure to high glucose for 24 h (1.4-fold, *p* < 0.05) (Figure 1D,G) and 48 h (1.9-fold, *p* < 0.05) (Appendix A). However, co-immunoprecipitation studies showed a reduction in the association between BRCA1 and phosphorylated ACCA (*p* < 0.001) (Figure 1H,I). ACCA is an exclusively cytoplasmic protein, whereas BRCA1 is a nucleocytoplasmic-shuttling protein found in the cytoplasm and nucleus [4,37,38,39]. We therefore wondered whether the reduced association between BRCA1 and phosphorylated ACCA may be due to altered localisation of BRCA1. Subcellular fractionation studies revealed that BRCA1 was predominantly cytoplasmic in both normal and high glucose conditions (Figure 1J). Taken together, the data suggest that high glucose reduced the association between BRCA1 and p-ACCA (S^79^) and upregulated FASN, resulting in a lipogenic phenotype.

### 2.2. Glucose Level Regulates Key Proteins Involved in Endogenous Fatty Acid Synthesis in T47D Breast Cancer Cells

Although ACCA phosphorylation paralleled ACCA abundance as observed in MCF7 cells, both ACCA phosphorylation and abundance reduced after 24 h in high glucose in T47D cells (Figure 2A,B). However, results similar to MCF7 cells were obtained after 48 h; both ACCA phosphorylation and total ACCA increased in high glucose, 1.3-fold and 1.5-fold (*p* < 0.05), respectively (Figure 2A,C). Similarly, when ACCA phosphorylation was normalised to total ACCA, we observed no difference in normalised ACCA phosphorylation between normal and high glucose (Figure 2B,C). Downstream of ACCA, FASN abundance increased slightly in high glucose (Figure 2D,E and Appendix A). Similar to MCF7 cells, AMPK phosphorylation declined in high glucose after 24 h (*p* < 0.05) (Figure 2D,F) and 48 h (*p* < 0.01) (Appendix A). In contrast to MCF7 cells however, BRCA1 abundance was lower in high glucose (*p* < 0.05) (Figure 2D,G and Appendix A). Additionally, the association between BRCA1 and p-ACCA (S^79^) was increased in high glucose (Figure 2H,I) and this is opposite to what we observed in MCF7 cells. This increase in the association between BRCA1 and p-ACCA (S^79^) was not due to BRCA1 shuttling, since BRCA1 was predominantly cytoplasmic in both normal and high glucose (Figure 2J).

### 2.3. Normal Glucose Blocks an IGF-I-Induced Lipogenic Phenotype in T47D and MCF7 Breast Cancer Cells

We next studied the impact of normal glucose levels on the lipogenic actions of IGF-I. MCF7 and T47D breast cancer cells were treated with increasing doses of IGF-I under normal and high glucose conditions. Twenty-four-hour treatment with IGF-I induced ACCA dephosphorylation in a dose-dependent manner in MCF7 cells under high glucose conditions as previously reported, consistent with ACCA activation (Figure 3A,B). Under normal glucose conditions however, there was a dose-dependent increase in ACCA phosphorylation in response to IGF-I, suggesting that ACCA was deactivated. To assess the impact of glucose level on IGF-I-induced upregulation of FASN protein, cells were dosed with IGF-I for 48 h under normal and high glucose conditions. Consistent with our previous findings, IGF-I upregulated FASN abundance in high glucose conditions (Figure 3A,C). In contrast, normal glucose conditions completely ablated this effect, resulting in no increase in FASN abundance in response to IGF-I. These observations were also confirmed in the T47D breast cancer cell line (Figure 3D–F). Collectively, the data suggest that in contrast to high glucose, normal glucose blocks an IGF-I-induced lipogenic phenotype in MCF7 and T47D breast cancer cells.

### 2.4. Normal Glucose Promotes BRCA1 Function of Suppressing an IGF-I-Induced Lipogenic Phenotype in MCF7 and T47D Breast Cancer Cells

Having shown that IGF-I-induced ACCA dephosphorylation is inhibited by normal glucose, we next sought to investigate whether this effect is mediated via alteration of the association between BRCA1 and p-ACCA (S^79^). MCF7 cells were first cultured in normal glucose medium for 24 h and serum-starved for another 24 h in normal glucose serum-free media, before short-term treatment with IGF-I for 30 min in serum-free media containing normal or high glucose. Short-term treatment with IGF-I was chosen to avoid confounding effects of transcriptional regulation of protein abundance. Western blotting analysis confirmed that 30 min treatment of T47D cells with IGF-I in normal and high glucose recapitulates the phenotype observed with 24 h treatment, where normal glucose inhibited IGF-I-induced ACCA dephosphorylation and FASN upregulation (Figure 4A,B). Immunoprecipitation studies showed that normal glucose blocked IGF-I-induced ACCA dephosphorylation by increasing the association between BRCA1 and p-ACCA (S^79^) and these findings are completely opposite of those observed in high glucose (Figure 4C). The same phenotype was confirmed in MCF7 cells, showing that IGF-I induces ACCA dephosphorylation by reducing the association between BRCA1 and p-ACCA (S^79^) but only in high glucose (Figure 4D–G). Collectively, the data suggest that normal glucose supports regulation of the lipogenic action of IGF-I by BRCA1.

### 2.5. High Glucose Modulates IGF-I-Induced Cell Growth in Breast Cancer Cells, but Not in Benign Mammary Epithelial Cells

To assess the impact of glucose on IGF-I-induced proliferation, normal and malignant breast cell lines were exposed to normal and high levels of glucose and were stimulated with IGF-I (0–50 ng/mL). In MCF7 breast cancer cells, IGF-I induced proliferation under both normal and high glucose conditions, however, the cells showed a higher and sustained response to IGF-I in high glucose compared to normal glucose (Figure 5A). T47D breast cancer cells did not show any response to IGF-I in normal glucose while the proliferative response under high glucose conditions was increased at 10 and 25 ng/mL IGF-I (*p* < 0.05), but declined to the level observed in normal glucose at 50 ng/mL (Figure 5B). In contrast to the cancer cells, there was no difference in the proliferative response to IGF-I between normal and high glucose in normal mammary epithelial cells MCF10A cells (Figure 5C). When compared to normal glucose, high glucose alone did not significantly increase the rate of proliferation in malignant or non-malignant cells, whereas addition of IGF-I (25 ng/mL) increased the rate of proliferation in all the cells lines except for T47D cells in normal glucose (Figure 5D). However, high glucose significantly increased the effect of IGF-I in cancer cells compared to normal cells, in which glucose level did not have an impact on the proliferative effect of IGF-I.

## 3. Discussion

We have shown previously in breast cancer cells cultured in high glucose that IGF-I reduced the interaction between BRCA1 and phosphorylated ACCA [16] and that it induced cell growth in a FASN-dependent manner [40]. Here, we studied these IGF-I actions in normal physiological glucose conditions and we report that normal glucose blocked these IGF-I lipogenic actions in MCF7 and T47D breast cancer cells. First, we studied the impact of glucose levels on the abundance of lipogenic enzymes in the absence of any exogenous growth factor. Although ACCA is regulated by reversible phosphorylation, net ACCA phosphorylation was not altered by glucose levels, possibly due to concurrent alteration of total ACCA abundance. Concurrent reduction of ACCA phosphorylation and total ACCA abundance has previously been shown to result in no net change in normalised ACCA phosphorylation, ACCA enzymatic activity, and lipid synthesis in HCT-8 colorectal cancer cells [41]. ACCA regulation involves a complex interplay among reversible phosphorylation, allosteric regulation by substrates and products, and gene expression subject to metabolic status, as well as protein–protein interactions [42,43]. Phosphorylation serves as a short-term regulatory mechanism which directly inhibits ACCA activity, while long-term regulation involves gene transcription [43]. Our results may therefore reflect a combination of both acute and chronic ACCA regulation, where upregulated transcription of the *ACACA* gene produces newly synthesized enzyme molecules in a phosphorylated and inactive state. In addition, the data suggest that other phosphoregulators of ACCA besides AMPK such as phosphokinase C and BRCA1 may also be involved [15,39,44]. 

BRCA1 affects fatty acid synthesis by interacting with the phosphorylated and inactive form of ACCA, preventing p-ACCA dephosphorylation and activation [15,39]. We observed a reduction in the association between BRCA1 and p-ACCA (S^79^) in high glucose, despite an increase in BRCA1 and p-ACCA (S^79^) protein abundance in MCF7 cells. In contrast, BRCA1 and p-ACCA (S^79^) abundance were reduced by high glucose in T47D cells, yet the association between them was increased. The reason for these puzzling observations is unknown, but they support the evidence we previously published showing that the association between BRCA1 and p-ACCA (S^79^) is independent of p-ACCA (S^79^) abundance [16]. It is possible that cellular energy status plays a role in regulating this association, considering that MCF7 and T47D cells display significant bioenergetic differences. Energy metabolism in MCF7 is predominantly oxidative compared to T47D cells which are more glycolytic and convert more pyruvate to lactate [45,46]. Higher mitochondrial activity and efficiency in MCF7 cells, in the context of high glucose, generates more metabolic intermediates in the mitochondria which are precursors for fatty acid synthesis. Therefore, the reduced association between BRCA1 and p-ACCA (S^79^) under high glucose conditions in MCF7 cells is permissive for increased fatty acid synthesis. Although T47D cells are more glycolytic than MCF7 cells, it is possible that they can also switch to predominantly oxidative metabolism under normal physiological glucose conditions to produce mitochondrial intermediates for fatty acid synthesis. Mutant *p53* such as present in T47D cells [47], has been shown to contribute to the bioenergetic differences [48], however, alternative p53-independent pathways may also be activated in the absence of functional p53. Stimulation of mitochondrial respiration during glucose deprivation has been demonstrated in HeLa cells, suggesting that cancer cells can switch between glycolytic and oxidative metabolism [30,49,50]. Given the complex role of p53 in metabolism, more work is required to fully understand its role in the current model. Additionally, future work will also explore the impact of long-term exposure to high glucose on the association between BRCA1 and p-ACCA (S^79^).

Our current data extend our previous findings on the role of IGF-I in the fatty acid synthesis pathway. Previously, we found that IGF-I induces ACCA dephosphorylation by reducing the association between BRCA1 and p-ACCA (S^79^) [16], thereby rendering ACCA susceptible to activation. While our previous experiments were conducted in high glucose concentrations (25 mM), as conventionally used in cell culture, physiological glucose levels are usually far lower, ranging from 5 to 7 mM [33]. In the current study, we showed that lipogenic effects of IGF-I under high glucose conditions were completely blocked by lowering glucose to physiological levels. ACCA phosphorylation desensitizes ACCA to activation [51], therefore ACCA hyper-phosphorylation in response to IGF-I in normal physiological glucose is consistent with ACCA inhibition. However, these results would have to be confirmed by actual measurement of ACCA activity. ACCA inhibition by pharmacological inhibitors that mimic ACCA phosphorylation have been found to reduce fatty acid synthesis, suppress hepatocellular carcinoma, and improve metabolic function [52,53]. While IGF-I induced ACCA dephosphorylation by reducing the association between BRCA1 and phosphorylated ACCA in high glucose, this was not observed under normal physiological glucose conditions. The current data suggest that BRCA1 prevented IGF-I-induced ACCA dephosphorylation in normal physiological glucose, raising the possibility that maintaining normal glucose levels supports BRCA1 function and that it is functionally equivalent to ACCA inhibition.

Under high glucose conditions, IGF-I induced a lipogenic phenotype by increasing FASN abundance downstream of ACCA and that ER-positive breast cancer cells were dependent on FASN upregulation for their proliferative response to IGF-I [16,40]. In the current study, normal physiological glucose completely blocked FASN upregulation and impaired the proliferative response of ER-positive breast cancer cells to IGF-I. Since BRCA1 has also been shown to negatively regulate FASN [16], our data suggest that normal physiological glucose conditions are critical for BRCA1 function in restraining IGF-I-induced upregulation of FASN and the associated proliferative responses. In normal mammary epithelial cells however, we observed no difference in the proliferative response to IGF-I under normal and high glucose conditions. This observation may reflect the low endogenous fatty acid synthesis observed in normal cells [54] and are consistent with our previous data showing that the proliferative response to IGF-I was not FASN-dependent in these cells [55]. In addition, BRCA1 is predominantly nuclear in normal mammary epithelial cells [16], suggesting that the association between BRCA1 and p-ACCA in the cytoplasm regulates ACCA only in cancer cells which have upregulated fatty acid synthesis. In both normal and breast cancer cell lines, the proliferative response to IGF-I declined at the highest concentration used (50 ng/mL). This may be due to induction of phosphatase and tensin homolog (PTEN) which has been shown to block response to higher doses of insulin-like growth factor II (IGF-II) [56]. Targeting either *ACCA* or *FASN* genes in breast cancer cells using siRNA significantly suppressed fatty acid synthesis, induced apoptosis and arrested cell proliferation, suggesting that ACCA and FASN are equally essential for cell survival [57].

We acknowledge that high glucose and serum-starvation in our study may cause cell stress [58,59], possibly resulting in independent outcomes unrelated to glucose homeostasis. In addition, IGF-I has been previously linked with correction of mitochondrial dysfunction [60,61], bringing into question whether our current findings also reflect potential mitoprotective effects of IGF-I. However, we observed similar effects of IGF-I between T47D cells and MCF7 cells, despite T47D cells displaying more mitochondrial dysfunction relative to MCF7 cells [46], making such a possibility less likely. We conducted our assays under serum-starvation and high glucose conditions in order to make direct comparison with our previously reported findings obtained in serum-starved cells under high glucose conditions [16]. Oxidative stress induced by high glucose has been reported to impair cell proliferation in MCF7 cells [62], however, high glucose alone did not affect cell proliferation in our study. In a separate study, our group has previously reported in MCF7 that high glucose induced the Warburg effect, as characterized by an increase in lactate secretion, together with increased lactate transporters and reduced oxidative phosphorylation [32]. Endogenous, glucose-derived lactate may increase transcription of oncogenes and tumour suppressors such as BRCA1 [63] and may be preferred over glucose by tumour cells for oxidative metabolism [64]. While these possible independent effects may not be completely accounted for in the current study, spent media was changed every 24 h to avoid accumulation of metabolism by-products such as lactate [65].

We propagated our cells in growth media supplemented with serum and conducted the assays under serum starvation conditions. Although serum provides optimal conditions for propagating cultured cells [66], it is a poorly-defined mixture of growth factors, hormones, and other elements, with lot-to-lot variation [67,68]. We therefore performed experiments under serum-free conditions to ensure reproducible experimental conditions and to eliminate potential confounding effects by hormones and growth factors akin to IGF-I. To minimize the impact of serum-starvation, we supplemented serum-free media with sodium bicarbonate, apo transferrin, glutamine, and albumin, which are some components of serum [66,68].

To the best of our knowledge, the impact of glucose on the BRCA1-ACCA association and on the effect of IGF-I on this complex has not been reported previously. Our data in high glucose conditions, which we also previously reported, suggested that IGF-I circumvents tumour suppressive functions of BRCA1 by reducing its association with phosphorylated ACCA and upregulating FASN downstream of ACCA (Figure 6A). This was not observed under normal glucose conditions in our current study. With our data, we propose a model in which normal physiological glucose conditions support BRCA1 functions in restraining lipogenic actions of IGF-I to regulate cell growth and proliferation (Figure 6B).

## 4. Materials and Methods

### 4.1. Reagents

Recombinant, human IGF-I peptide was purchased from Gropep Bioreagents (Adelaide, South Australia, Australia).

### 4.2. Cell Culture

Human breast cancer cells MCF-7 and T47D were purchased from the European Collection of Authenticated Cell Cultures (ECACC, Salisbury, UK). MCF10A, a non-tumorigenic epithelial cell line was purchased from the American Type Culture Collection (ATCC, Manassas, VA, USA). All the cells were authenticated by short tandem repeat (STR) profiling and were used at passage 20 or below. The cells were grown in a humidified 5% carbon dioxide (CO_2_) atmosphere at 37 °C. MCF-7 and T47D cells were cultured in DMEM (Sigma Aldrich Corp., St. Louis, MO, USA) with 25 mM glucose, supplemented with 10% foetal bovine serum (Gibco by Life Technologies, Carlsbad, CA, USA), and 2 mM l-glutamine (Lonza, Manchester, UK). MCF-10A were maintained in 1:1 mixture of Ham’s F12 medium and DMEM with 2.5 mM l-glutamine (Gibco by Life Technologies, Carlsbad, CA, USA) and 19.5 mM Glucose, supplemented with 5% horse serum (Gibco by Life Technologies, Carlsbad, CA, USA), 20 ng/mL epidermal growth factor (Calbiochem, Nottingham, UK), 100 ng/mL cholera toxin (Sigma Aldrich Corp., St. Louis, MO, USA), 10 μg/mL insulin (Novo Nordisk, West Sussex, UK), and 0.5 μg/mL hydrocortisone (Sigma Aldrich Corp., St. Louis, MO, USA). For experiments, cells were first cultured in DMEM containing 5 mM glucose and allowed to adhere to culture vessels for 24 h. The media was then replaced with serum-free growth media containing either 5 or 25 mM glucose for 24 or 48 h. For IGF-I treatments, cells were serum-starved in serum-free media for 24 h before treatment with IGF-I for 24 or 48 h.

### 4.3. Western Blotting

Cells were either lysed directly on culture plates and scraped or trypsinised first before lysis using cell lysis buffer described elsewhere [69]. Equal amount of proteins (15–40 µg) from lysates were separated on 6–8% sodium dodecyl sulfate-polyacrylamide gel electrophoresis (SDS–PAGE) at 140 V for 1 h 30 min and transferred onto supported nitrocellulose membranes (Bio-Rad, Hertfordshire, UK) at 100 V for 1 h. Non-specific binding was eliminated by incubating membranes with 1% *w*/*v* milk in tris-buffered saline Tween-20 (TBST) for 1 h at room temperature. Membranes were then probed overnight at 4 °C with the following antibodies following manufactures’ instructions; acetyl CoA carboxylase (ACCA) (1:1000, Cell Signalling, Danvers, MA, USA), phospho-ACCA (S^79^) (1:1000, Cell Signalling, Danvers, MA, USA), AMP-activated protein kinase (AMPK) (1:1000, Cell Signalling, Danvers, MA, USA), p-AMPK (1:1000, Cell Signalling, Danvers, MA, USA), fatty acid synthase (FASN) (1:1000, BD Biosciences, Wokingham, Berkshire, United Kingdom), breast cancer 1, early onset (BRCA1) (1:500, Merck Millipore, Billerica, MA, USA), glyceraldehyde 3-phosphate dehydrogenase (GAPDH) (1:5000, Merck Millipore, Billerica, MA, USA), and tubulin (1:5000, Merck Millipore, Billerica, MA, USA). Membranes were washed and incubated with specific peroxidase-conjugated secondary antibodies. Membranes were washed with TBST and the signal was detected using Super Signal West Dura and Femto Chemiluminescent Substrates (Thermo Scientific, Rockford, IL, USA). Images were acquired with a ChemiDoc MP System imaging system equipped with a cooled CCD camera and Image Lab Software 5.2.1 (Bio-Rad Laboratories Ltd., Hemel Hempstead, UK). The bands were analysed using Image J 1.6.0 65 (National Institutes of Health, Bethesda, MD, USA).

### 4.4. Cell Counting

Cells were seeded in 6-well plates at 0.2 × 10^6^ cells per well in growth media containing normal glucose (5 mM) and maintained for 24 h in 5% CO_2_ at 37 °C. Growth media was changed to serum-free media containing either low (5 mM) or high glucose (25 mM) and cells were serum-starved for 24 h. Cells were then treated with 0, 10, 25, and 50 ng/mL IGF-I in serum-free medium containing low or high glucose for 24 h. Each IGF-I dose was performed in triplicate. Cells were then trypsinised and counted using the trypan blue exclusion method as previously described [69].

### 4.5. Co-Immunoprecipitation

Cells were lysed in cell lysis buffer supplemented with protease inhibitors and phosphatase inhibitor cocktail (Sigma-Aldrich, St. Louis, MO, USA). Equal amounts of protein (1 mg) from the resulting lysates were incubated with 2 µg of anti-BRCA1 (Merck Millipore, Billerica, MA, USA), anti-p-ACCA (S^79^) (Cell Signalling, Danvers, MA, USA), and negative control IgG (Dako, Glostrup, Denmark) antibodies overnight at 4 °C. Lysates were then incubated with 10 μg affinity purified secondary antibody (Merck Millipore, Billerica, MA, USA) for 1 h, followed by incubation with Protein A/G PLUS immunoprecipitation reagent (Santa Cruz, Dallas, TX, USA) for 1 h at 4 °C. After centrifugation and washing with lysis buffer, immune complexes were eluted by heating with sample loading buffer (Sigma-Aldrich, St. Louis, MO, USA) for 5 min at 95 °C and analysed by Western blotting.

### 4.6. Subcellular Fractionation

Extraction of cytoplasmic and nuclear protein fractions was achieved using a NE-PER Nuclear and Cytoplasmic Extraction Kit (Thermo Scientific, Rockford, IL USA). Whole cell fractions were obtained by routine whole cell lysis using cell lysis buffer. Protein concentrations were determined using Pierce BCA Protein Assay (Thermo Scientific, Rockford, IL USA) and equal amounts of cytosolic, nuclear, and whole cell extracts were analysed by Western blotting.

### 4.7. Statistical Analyses

Data were analysed using IBM SPSS statistics 23, v. 23.0.0.2 (IBM Corporation, Portsmouth, Hampshire, UK) and presented as mean ± SEM of a minimum of three independent experiments. Independent samples *t*-test was used to compare means of two groups and for the means of more than two groups, one-way analysis of variance (ANOVA) was used followed by a Bonferroni multiple comparison test. For both tests, a *p*-value of equal or less than 0.05 was considered statistically significant.

## Figures and Tables

**Figure 1 ijms-21-08674-f001:**
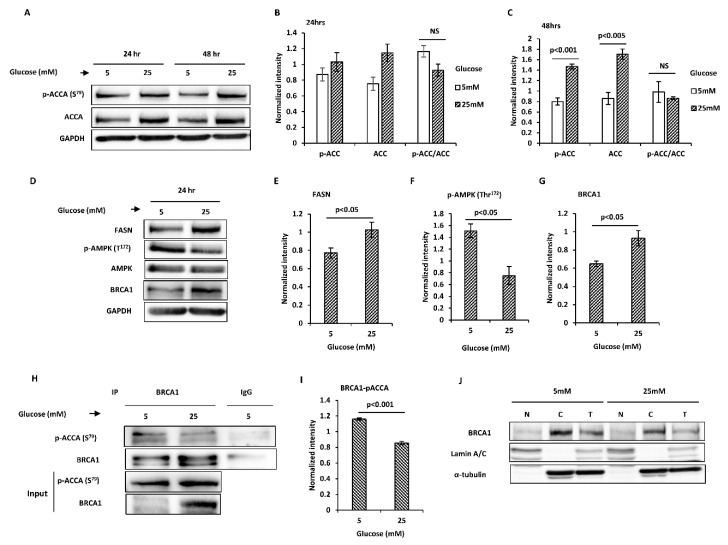
High glucose regulates key lipogenic proteins by glucose in MCF7 breast cancer cells. Cells were exposed to normal and high glucose for 24 and 48 h. (**A**) Representative immunoblots of total acetyl CoA carboxylase (ACCA) and p-ACCA (S^79^) are shown. (**B**,**C**) Densitometry analysis of total ACCA normalised to glyceraldehyde 3-phosphate dehydrogenase (GAPDH) and p-ACCA (S^79^) normalised to both GAPDH and total ACCA. (**D**) Representative immunoblots of fatty acid synthase (FASN), total AMP-activated protein kinase (AMPK), p-AMPK (Thr^172^) and breast cancer 1, early onset (BRCA1) are shown. (**E**–**G**) Densitometry analysis of p-AMPK (Thr^172^) normalised to total AMPK and FASN and BRCA1 normalised to GAPDH. (**H**) Lysates from cells exposed to normal and high glucose for 24 h were subjected to immunoprecipitation with 2 μg anti-BRCA1 antibody or negative control IgG, followed by Western blotting using BRCA1 and p-ACCA (S^79^) antibodies. Input represent lysates not used for immunoprecipitation. (**I**) The graph shows signal of precipitated p-ACCA (S^79^) normalised to BRCA1. (**J**) Western blot analysis of cytoplasmic and nuclear fractions from cells exposed to low and high glucose for 24 h. Lamin A/C and α-tubulin were used as nuclear and cytoplasmic markers, respectively. All results shown are representative of three independent experiments and the graphs represent mean ± SEM. Independent samples *t*-test was used to compare means of 2 groups.

**Figure 2 ijms-21-08674-f002:**
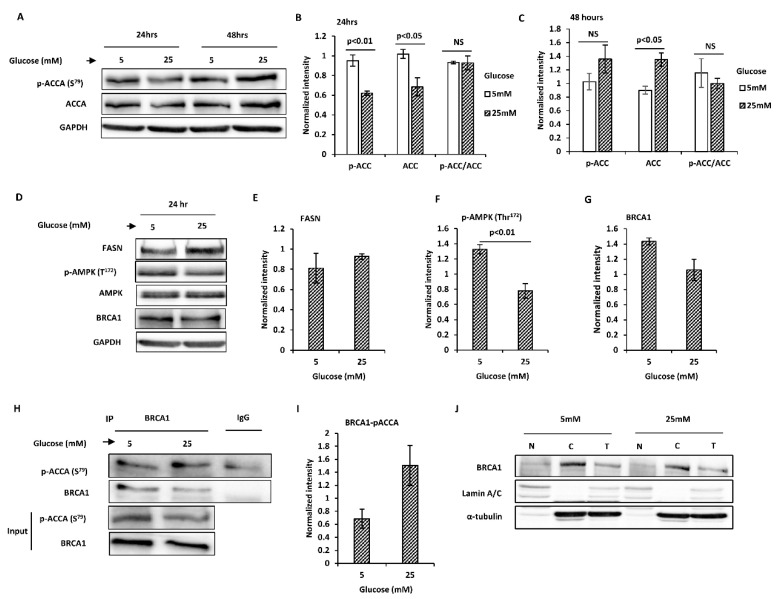
High glucose regulates key lipogenic proteins by glucose in T47D breast cancer cells. Cells were exposed to normal and high glucose for 24 and 48 h and whole cell lysates were subjected to Western blot analysis. (**A**) Representative blots of total acetyl CoA carboxylase (ACCA) and p-ACCA (S^79^) are shown. (**B**,**C**) Densitometry analysis of total ACCA normalised to glyceraldehyde 3-phosphate dehydrogenase (GAPDH) and p-ACCA (S^79^) normalised to both GAPDH and total ACCA. (**D**) Representative blots of fatty acid synthase (FASN), total AMP-activated protein kinase (AMPK), p-AMPK (Thr^172^) and breast cancer 1, early onset (BRCA1) are shown. (**E**–**G**) Densitometry analysis of p-AMPK (Thr^172^) normalised to total AMPK and FASN and BRCA1 normalised to GAPDH. (**H**) Lysates from cells exposed to normal and high glucose for 24 h were subjected to immunoprecipitation with 2 μg anti-BRCA1 antibody or negative control IgG, followed by Western blotting using BRCA1 and p-ACCA (S^79^) antibodies. Input represent lysates not used for immunoprecipitation. (**I**) The graph shows signal of precipitated p-ACCA (S^79^) normalised to BRCA1. (**J**) Western blotting of cytoplasmic and nuclear fractions from cells exposed to low and high glucose for 24 h. Lamin A/C and α-tubulin were used as nuclear and cytoplasmic markers, respectively. All results shown are representative of three independent experiments and the graphs represent mean ± SEM. Independent samples *t*-test was used to compare means of 2 groups. High glucose regulates key lipogenic proteins by glucose in T47D breast cancer cells. Cells were exposed to normal and high glucose for 24 and 48 h and whole cell lysates were subjected to Western blot analysis. (**A**) Representative blots of total acetyl CoA carboxylase (ACCA) and p-ACCA (S^79^) are shown. (**B**,**C**) Densitometry analysis of total ACCA normalised to glyceraldehyde 3-phosphate dehydrogenase (GAPDH) and p-ACCA (S^79^) normalised to both GAPDH and total ACCA. (**D**) Representative blots of fatty acid synthase (FASN), total AMP-activated protein kinase (AMPK), p-AMPK (Thr^172^) and breast cancer 1, early onset (BRCA1) are shown. (**E**–**G**) Densitometry analysis of p-AMPK (Thr^172^) normalised to total AMPK and FASN and BRCA1 normalised to GAPDH. (**H**) Lysates from cells exposed to normal and high glucose for 24 h were subjected to immunoprecipitation with 2 μg anti-BRCA1 antibody or negative control IgG, followed by Western blotting using BRCA1 and p-ACCA (S^79^) antibodies. Input represent lysates not used for immunoprecipitation. (**I**) The graph shows signal of precipitated p-ACCA (S^79^) normalised to BRCA1. (**J**) Western blotting of cytoplasmic and nuclear fractions from cells exposed to low and high glucose for 24 h. Lamin A/C and α-tubulin were used as nuclear and cytoplasmic markers, respectively. All results shown are representative of three independent experiments and the graphs represent mean ± SEM. Independent samples *t*-test was used to compare means of 2 groups.

**Figure 3 ijms-21-08674-f003:**
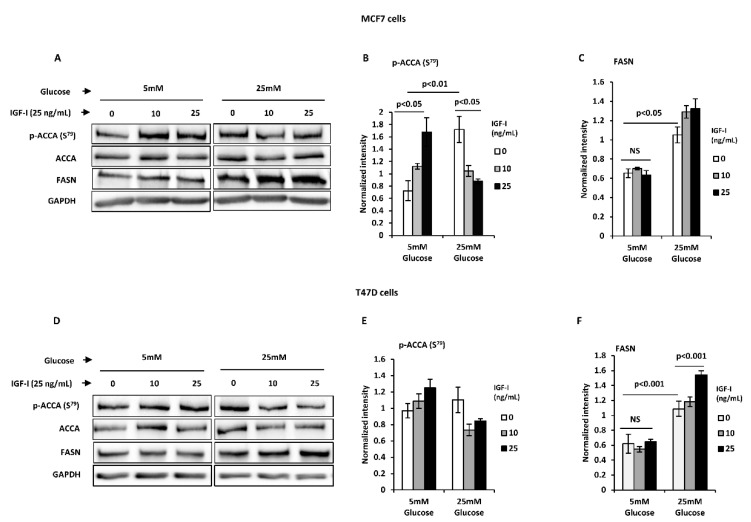
High glucose favours insulin-like growth factor I (IGF-I)-induced lipogenic phenotype in T47D and MCF7 breast cancer cells. Western blot analysis of (**A**) MCF7 and (**D**) T47D cells treated with 0, 10, and 25 ng/mL IGF-I for 24 h under normal and high glucose conditions. Representative blots of each protein from at least three experiments are shown. The graphs show densitometry analyses of phosphorylated acetyl CoA carboxylase (p-ACCA) (S^79^) normalised to total ACCA and fatty acid synthase (FASN) normalised to glyceraldehyde 3-phosphate dehydrogenase (GAPDH) in (**B**,**C**) MCF7 cells and (**E**,**F**) T47D cells. Each bar represents mean ± SEM of three independent experiments. Statistical analyses were performed by one-way ANOVA, followed by Bonferroni’s multiple comparison test.

**Figure 4 ijms-21-08674-f004:**
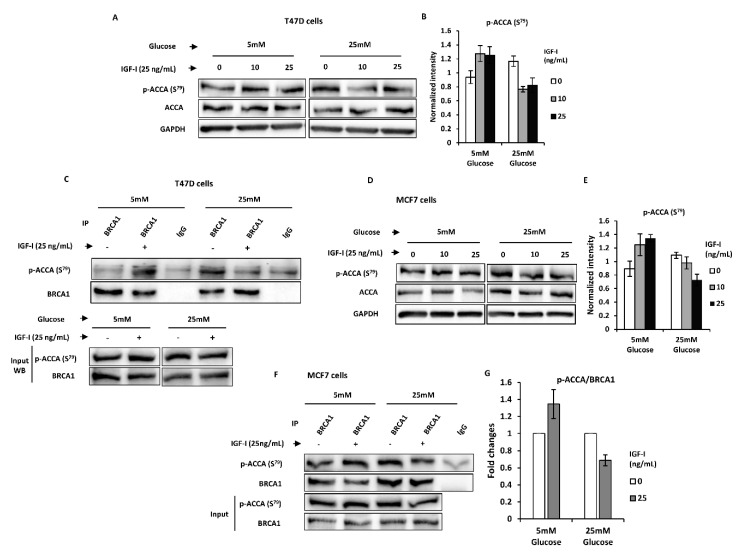
High glucose favours insulin-like growth factor I (IGF-I)-induced lipogenic phenotype by impairing breast cancer 1, early onset (BRCA1) function in MCF7 and T47D breast cancer cells. (**A**–**C**) T47D and (**D**–**G**) MCF7 cells were stimulated with 0, 10, and 25 ng/mL of IGF-I for 30 min. Lysates were analysed with Western blotting and (**A**,**D**) representative blots from at least 3 independent experiments are shown. (**B**,**E**) The graphs show densitometry analyses of phosphorylated acetyl CoA carboxylase (p-ACCA) (S^79^) normalised to total ACCA and each bar represents mean ± SEM. (**C**,**F**) Lysates from cells treated with 25 ng/mL IGF-I were immunoprecipitated with 2 μg anti-BRCA1 antibody or negative control IgG, followed by Western blotting using BRCA1 and p-ACCA (S^79^) antibodies. Inputs represent lysates not used for immunoprecipitation. (**G**) The graphs show densitometry analyses of immunoprecipitation reactions of p-ACCA (S^79^) normalised to BRCA1 and expressed as fold changes.

**Figure 5 ijms-21-08674-f005:**
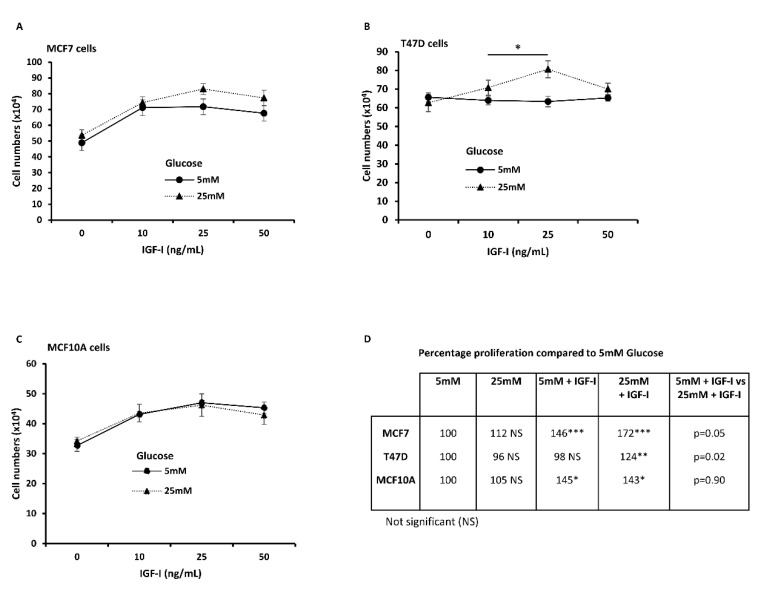
High glucose modulates insulin-like growth factor I (IGF-I)-induced cell growth in breast cancer cells, but not in benign mammary epithelial cells. Breast cancer cell lines; (**A**) MCF7, (**B**) T47D, and normal mammary epithelial cell line (**C**) MCF10A were treated with 0, 10, and 25 ng/mL IGF-I for 48 h under normal and high glucose conditions. Total cell numbers were determined using trypan blue exclusion dye cell counting. The graphs represent mean ± SEM of three independent experiments with each condition performed in triplicate. (**D**) Absolute cell numbers after exposure to normal and high glucose with or without 25 ng/mL, normalised to normal glucose (5 mM) and expressed as a percentage. Three independent experiments were performed, and mean values are presented. One-way ANOVA, followed by Bonferroni’s multiple comparison test was used statistical analyses. * *p* < 0.05, ** *p* < 0.01 and *** *p* < 0.001.

**Figure 6 ijms-21-08674-f006:**
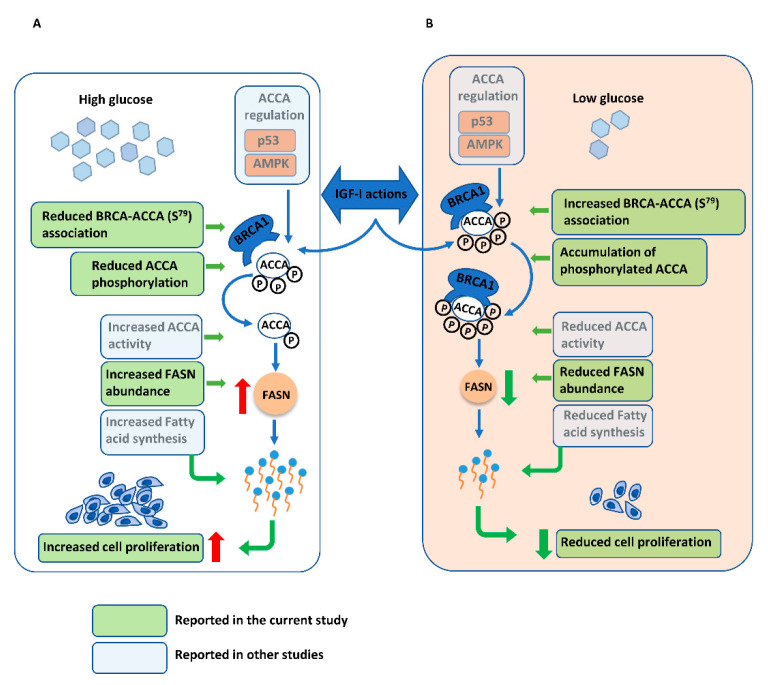
Proposed model of how normal glucose supports breast cancer 1, early onset (BRCA1) function as a metabolic restraint of insulin-like growth factor I (IGF-I) actions. Acetyl CoA carboxylase (ACCA) catalyses a rate-limiting step of the fatty acid synthesis pathway and is partly regulated at a transcriptional level by p53 and post transcriptionally by AMP-activated protein kinase (AMPK) via reversible phosphorylation [15,41]. Its activity depends on the equilibrium between active (dephosphorylated) and less active (phosphorylated) forms [43] and the novel BRCA1 functions in regulating fatty acid synthesis involve binding phosphorylated ACCA and keeping it in this less active form. (**A**) In high glucose conditions, IGF-I impairs this function by reducing the association between BRCA1 and p-ACCA (S^79^), resulting in ACCA dephosphorylation and increased activity. Downstream of ACCA, IGF-I increases fatty acid synthase (FASN) abundance, leading to increased fatty acid synthesis to support cell proliferation. (**B**). In contrast, the association between BRCA1 and p-ACCA (S^79^) increases in response to IGF-I in normal glucose conditions, resulting in increased ACCA phosphorylation and reduced activity, suggesting that normal glucose support BRCA1 function. Downstream of ACCA, normal glucose conditions inhibit IGF-I-induced upregulation of FASN and consequently, the proliferative responses to IGF-I are reduced. While our conclusions are based on changes in protein abundance and protein–protein interactions, alterations in the fatty acid synthesis pathway would have to be confirmed by actual measurement of enzyme activity or lipids.

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
