# Peer review of "Glucose Concentration in Cell Culture Medium Influences the BRCA1-Mediated Regulation of the Lipogenic Action of IGF-I in Breast Cancer Cells"

_ijms, 2020, doi:10.3390/ijms21228674_

Round 1

Reviewer 1 Report

The manuscript by Koobotse et al. is a revision of previously reviewed documents. Major improvements have been made. A sticking point is the use of “normal”, especially in the title. The authors are invited to consider the following suggestion:

In the Materials and Methods section:

Line 412: The cells were maintained as previously described [37].

Instead of referring to [37] (Zheng Endocrine-Related Cancer. 2010;17:539–551), provide a better description of the culture conditions/media. Does the mention in [37] that the medium was supplemented with L-glucose (in order to exclude confounding effects not mediated by enhanced glucose metabolism, such as osmotic effects) still apply?

A clearer section in the Materials and Methods would also allow a clearer transition between glucose in blood and glucose in cell culture in the introduction. A suggestion would be to add the following sentences (or variation of) after line 81: 

Glucose is a soluble hexose sugar added to cell culture media in amounts ranging from 1g/L (5.5 mM) to as high as 10 g/L (55 mM). Supplementation with approximately 5.5 mM D-glucose approximates normal blood sugar levels in vivo, while concentrations of glucose around 10mM and above 10mM are analogous to pre-diabetic and diabetic levels respectively. Although high glucose concentrations are known to have detrimental effects on many cell types, it is not uncommon for media to contain diabetic levels of glucose supplementation, nor for scientists to study the effect of glucose levels (ex: PMID: 28228375, 19386985, 31752674, 23329407…)

Our previous study [16] was performed in [name of the culture medium] which was supplemented with xx mM of D-glucose (if L-glucose was used, indicate why…). Indicate whether this is conventional or not for the type of cells used in the study.

In the present study, we investigate the influence of glucose levels by comparing data generated when using either 5 mM (later referred as normal) or 25 mM (later referred as high) glucose in the cell culture medium. …

To avoid “normal” in the title, consider the following alternative title: The glucose concentration in the cell culture medium influences the regulation of the lipogenic action of IGF-I by BRCA1

Author Response

We submit our manuscript following minor revisions for further consideration for publication in the IJMS. We would like to apologize for a slight delay in our response and thank you so much for your patience with our submission. We thank the reviewers for their constructive comments and suggestions which have helped improve the quality of the paper.

REVIEWER #1’S COMMENTS

  • The manuscript by Koobotse et al. is a revision of previously reviewed documents. Major improvements have been made. A sticking point is the use of “normal”, especially in the title. The authors are invited to consider the following suggestion:

We thank the reviewer for the comments on our revisions to date, and we appreciate their continued effort in reviewing our manuscript.  We understand the potential confusion between normal physiological levels of glucose and normal glucose levels used in cell culture experiments. The phrase “normal glucose” has been removed from the title. The title now reads:

“Glucose concentration in cell culture medium influences the BRCA1-mediated regulation of the lipogenic actions of IGF-I in breast cancer cells .”

  • In the Materials and Methods section:Line 412: The cells were maintained as previously described [37].Instead of referring to [37] (Zheng Endocrine-Related Cancer. 2010; 17:539–551), provide a better description of the culture conditions/media. Does the mention in [37] that the medium was supplemented with L-glucose (in order to exclude confounding effects not mediated by enhanced glucose metabolism, such as osmotic effects) still apply? 

           We thank the reviewer for the suggestion. The sentence, “The cells were maintained as previously described [37]”, was meant to avoid describing established method of culturing cells that has been previously described, i.e. humidified 5% CO2 atmosphere, 37oC and culture media with supplements that were previously described in Zheng Endocrine-Related Cancer. 2010; 17:539–551.

We acknowledge that citing a previous paper from our group inadvertently implied that other methods in the cited paper were applied in the current study. Although we previously used L-glucose as an osmotic control as described in Zeng et al, this does not apply in the current manuscript.

To provide more clarity, we have added a paragraph in the Materials and Methods section as follows:

Line 429:

The cells were grown in a humidified 5% carbon dioxide (CO2) atmosphere at 37oC.  MCF-7 and T47D cells were cultured in DMEM (Sigma Aldrich Corp., St. Louis, MO, USA) with 25mM glucose, supplemented with 10% foetal bovine serum (Gibco by Life Technologies, Carlsbad, CA, USA), and 2mM L-glutamine (Manchester, United Kingdom). MCF-10A were maintained in 1:1 mixture of Ham's F12 medium and DMEM with 2.5 mM l-glutamine (Gibco by Life Technologies, Carlsbad, CA, USA) with 19.5mM Glucose, supplemented with 5% horse serum (Gibco by Life Technologies, Carlsbad, CA, USA), 20 ng/ml epidermal growth factor (Calbiochem, Nottingham, UK), 100 ng/ml cholera toxin (Sigma Aldrich Corp., St. Louis, MO, USA), 10 μg/ml insulin (Novo Nordisk, West Sussex, UK) and 0.5 μg/ml hydrocortisone (Sigma Aldrich Corp., St. Louis, MO, USA). For experiments, cells were first cultured in DMEM containing 5mM glucose and allowed to adhere to culture vessels for 24 hours. The media was then replaced with serum-free growth media containing either 5mM or 25mM glucose for 24 or 48 hours. For IGF-I treatments, cells were serum-starved in serum-free media for 24 hours before treatment with IGF-I for 24 or 48 hours.

  • A clearer section in the Materials and Methods would also allow a clearer transition between glucose in blood and glucose in cell culture in the introduction. A suggestion would be to add the following sentences (or variation of) after line 81:Glucose is a soluble hexose sugar added to cell culture media in amounts ranging from 1g/L (5.5 mM) to as high as 10 g/L (55 mM). Supplementation with approximately 5.5 mM D-glucose approximates normal blood sugar levels in vivo, while concentrations of glucose around 10mM and above 10mM are analogous to pre-diabetic and diabetic levels, respectively. Although high glucose concentrations are known to have detrimental effects on many cell types, it is not uncommon for media to contain diabetic levels of glucose supplementation, nor for scientists to study the effect of glucose levels (ex: PMID: 28228375, 19386985, 31752674, 23329407…)Our previous study [16] was performed in [name of the culture medium] which was supplemented with xx mM of D-glucose (if L-glucose was used, indicate why…). Indicate whether this is conventional or not for the type of cells used in the study. In the present study, we investigate the influence of glucose levels by comparing data generated when using either 5 mM (later referred as normal) or 25 mM (later referred as high) glucose in the cell culture medium. … 

The reviewer makes a great suggestion and we have revised accordingly, with some modifications. We have added a paragraph to clarify the method in line 476 as described above

We have now added the suggested sentences with some modifications in line 83.

  • To avoid “normal” in the title, consider the following alternative title: The glucose concentration in the cell culture medium influences the regulation of the lipogenic action of IGF-I by BRCA1 

We thank the reviewer for this great suggestion. We have now changed the title, with some variation, to read; “Glucose concentration in cell culture medium influences the BRCA1-mediated regulation of the lipogenic actions of IGF-I in breast cancer cells .”

Reviewer 2 Report

This article has as main objective to evaluate impact of BRCA1 on metabolic activity and proliferative response to IGF-I anabolic actions in breast cancer cells cultured in normal glucose.

Although the findings are limited to in vitro experiments, the authors undelining with an  interesting sound the importants of glucose levels in controlling homeostatic processes and relative involvement of BRCA-I and ACCA in it.

However, the article is not easy to understand due to the various "flashbacks" on previous works hiding the present results which are interesting.

The manuscript need to be restructured focusing on present data, and avoiding speculation behond the in vitro enviroment.

Author Response

REVIEWER #2      

This article has as main objective to evaluate impact of BRCA1 on metabolic activity and proliferative response to IGF-I anabolic actions in breast cancer cells cultured in normal glucose. Although the findings are limited to in vitro experiments, the authors underlining with an interesting sound the important of glucose levels in controlling homeostatic processes and relative involvement of BRCA-I and ACCA in it.

However, the article is not easy to understand due to the various "flashbacks" on previous works hiding the present results which are interesting.

The manuscript needs to be restructured focusing on present data and avoiding speculation beyond the in vitro environment.

            We have submitted the previously reviewed manuscript with major revisions. In the original manuscript, reviewer 1 rightly suggested that we should downplay our conclusions on the effects of high glucose since these are not fully elucidated. Reviewer 1 further suggested a more appropriate way to present our data; describing that the conclusions from our previous study in 25 mM glucose are not observed when the cells are grown in normal conditions.

In the revised manuscript currently under review, Reviewer 2 has raised additional concerns regarding the clarity in the revisions we made to address concerns initially raised by Reviewer 1.

To address the current concerns, we have reduced the instances of “flashbacks “on previous works, only remaining with those that are relevant for context. We have made deletions at the following lines:

Line 192

Line 218

Line 323

Line 329

Line 345

To avoid speculation beyond in vitro environment, we have deleted a large paragraph at line 335, and made additional deletions at the following lines:

Line 364

Line 398

Line 418

This manuscript is a resubmission of an earlier submission. The following is a list of the peer review reports and author responses from that submission.

Round 1

Reviewer 1 Report

Review: Hyperglycaemia impairs BRCA1 in restraining IGF-1 in breast cancer cells

Background: A high energy supply promotes an unphosphorylated (active status) of acetyl-coA carboxylase (ACCA) and enhances endogenous lipogenesis which contributes to the tumorigenesis process. The functional interaction of BRCA1 with ACCA-driven endogenous lipogenesis has been studied for more than a decade. BRCA1 interacts with the phosphorylated (inactive) form of ACCA (p-ACCA), forming a complex that interferes with ACCA activity by preventing the dephosphorylation of pACCA. The interaction is abolished by several germline mutations in the BRCT domain of BRCA1: by allowing a constitutive active status of ACCA, they appear to mimic a high cellular energy supply, thus promoting enhanced cellular endogenous lipogenesis. ACCA is phosphorylated by upstream kinase AMPK. Treating BRCA1 mutation carriers with AMPK activators and or ACCA inhibitors was proposed in 2008 (PMID: 17620310 )

The authors have previously contributed to this field of study by showing that IGF-1, a potent mitogen of importance in the mammary gland, induced dephosphorylation of ACCA by reducing the interaction between BRCA1 and p-ACCA (PMID:30323899). In the present paper, they reconsider their data by investigating the consequences of growing cells in high glucose (analogous to diabetic conditions). Effects in two breast cancer cell lines (MCF-7 and T47D) are reported. The issue is worth looking at but major concerns are as follows:

  • The description of the culture conditions seems crucial yet it is not clear. Line 93: A paragraph on culture conditions for breast cancer cells seems desirable. The authors should clearly state the values of the glucose concentration in previous papers investigating the BRCA1-ACCA interaction. In the current version, it is only on line 289 that it is clearly stated that the previous experiments were conducted at 25mM glucose.
  • High glucose and serum-free media are stressful conditions for the cells. As reminded in PMID: 28228375, 25mM glucose may have independent effects on the outcome of the experiments not targeted to study of glucose homeostasis. This issue should be considered especially in view of the unexpected data that are observed (line 107).
  • Ray et al. have described a mechanism of regulation of ACCA activity distinct from the phosphorylation of Ser79 via AMPK (PMID: 19061860) . Why is this not considered? How does AMPK fit on the model proposed in Figure 6?
  • In PMID: 3289739 investigating the responsiveness of MCF-7 and T47D to IGF-1 , differences were reported. How do the authors justify using 10 and 25ng/ml for both cell lines?
  • When comparing MCF-7 and T47D, the authors report unexplained differences. The two cell lines have a different p53 status (wild-type/mutant). Considering that the IGF1 signaling axis and p53 are highly connected, how could this influence the data? How does p53 fit on the model proposed in Figure 6?
  • Molecular weight markers are not present on Figures. In figure 1H, the signal with BRCA1 appears as a doublet (especially at 25mM). What are these two bands?

Author Response

Reviewer 1

The description of the culture conditions seems crucial yet it is not clear. Line 93: A paragraph on culture conditions for breast cancer cells seems desirable. The authors should clearly state the values of the glucose concentration in previous papers investigating the BRCA1-ACCA interaction. In the current version, it is only on line 289 that it is clearly stated that the previous experiments were conducted at 25mM glucose.

We thank the reviewer for pointing this out. In the revised manuscript, we have provided more information about glucose concentration used in our previous study(page2,lines 80, 85 and86).

High glucose and serum-free media are stressful conditions for the cells. As reminded in PMID: 28228375, 25mM glucose may have independent effects on the outcome of the experiments not targeted to study of glucose homeostasis. This issue should be considered especially in view of the unexpected data that are observed (line 107).

The reviewerraised a valid point. We chose 25mM due to its widespread use as standard optimumconcentrationwidely-usedto culturebreast cancer cellsin vitro.Serum-starvationis necessary for cell synchronization to achieve ahomogenous population of growing cells and to remove potential confounding from poorly defined components of serum (Pirkmajer and Chibalin 2011, PMID: 28228375).In the absence of alternative approacheswithout caveats, we now acknowledge this as a limitation in the revised manuscript. We addedLine 354-357,page 10:We acknowledge that our data was obtained under conditions that cause cell stress, including high glucose and serum-starvation(Lamers et al 2011, PMID: 30092096)(Pirkmajer and Chibalin 2011, PMID: 28228375). While these approaches are necessary, in vitroto in vivoextrapolations always warrant caution.

Ray et al. have described a mechanism of regulation of ACCA activity distinct from the phosphorylation of Ser79 via AMPK (PMID: 19061860). Why is this not considered?

We thank the reviewer for pointing this out. We acknowledge that Ser1263 is a distinct phosphorylation sitefrom Ser79. While pSer1263 has been shown to facilitate the interaction between BRCA1 and pACCA, to the best of our knowledge, the relevance and impact of this interaction on the fatty acid synthesis pathway has not been demonstrated. The physiological relevance of this interactionhas only been linked to cell cycle regulation (Ray et al 2009, PMID 19061860). In contrast, the role of pSer79 in fatty acid synthesis has been previously well-described (Moreau et al, 2006, PMID: 16326698)(Fullerton, et al 2013, PMID:24185692)(Ortega et al, 2012, PMID: 22666314).

How does AMPK fit on the model proposed in Figure 6?

In figure 6, we propose a model showing how hyperglycaemia impacts BRCA1 function as a metabolic restraint of IGF-I actions. Our datasupports a model that that isdistinct from AMPK since the interaction between BRCA1 and pACCA is independent of the abundance of pACCAdownstream of AMPK. For example, in Figures 1H-I, the abundance of BRCA1 and pACCA arehigher in high glucose, but the BRCA1-pACCA associationis lower.

n PMID: 3289739 investigating the responsiveness of MCF-7 and T47D to IGF-1 , differences were reported. How do the authors justify using 10 and 25ng/ml for both cell lines?

We thank the reviewer for the comment and observation. In Karey et al 1988, PMID: 3289739, cells were cultured with IGF-I for 8 days with no media changes or addition(s). In our study, cells were only exposed to IGF-I for 24 to 48hrs, which therefore required higher doses. We also considered the minimum number of cells required toyield enough protein for subsequent protein analysis. Most importantly, we conducted dose response experiments [data not shown]that showed optimum responses at 10 and 25ng/ml. In our previously published studies(Koobotse et al, PMID: 30323899)(Zeng et al, 2010, PMID: 20356977) (Perks et al 2007, PMID: 17369847), we also used these doses.

When comparing MCF-7 and T47D, the authors report unexplained differences. The two cell lines have a different p53 status (wild-type/mutant). Considering that the IGF1 signaling axis and p53 are highly connected, how could this influence the data? How does p53 fit on the model proposed in Figure 6?

We acknowledge that we have reported differences in the impact of glucose in the BRCA1-pACCA interactionbetween MCF-7(wild-type p53)and T47D cells(mutant p53), in the absence of exogenous IGF-I. However, no differences were observed between the 2 cell lines in experiments that involved IGF-I, therefore it is difficult to describe how p53 would influence IGF-I results and how p53 would fit into the proposedmodel. Weproposed that the previously reported bioenergetic differences between MCF-7 and T47D cells (Radde et al 2015, PMID:25279503) may explain the differences we observed in the absence of exogenous IGF-I. In the revised manuscript, we have added: Line 290, page 9mutant p53, such asin T47D cells (Neve et al 2006, PMID: 17157791) has been shown to contribute to these bioenergeticdifferences(Kim et al 2019, PMID:30712844),however, alternative p53-independent pathways may also be activated in the absence of functional p53. Line 294, page 9Given the complex role of p53 in metabolism, more work is required to fully understand its role in the current model.Additionally...

Molecular weight markers are not present on Figures. In figure 1H, the signal with BRCA1 appears as a doublet (especially at 25mM). What are these two bands?

We thankthe reviewer for the observation and question. The upper strong band represents full length BRCA1(220KDa) and which is consistently present in all the BRCA1 blots in our experiments. The lower faint band (approx. 180-200KDa) may represent alternatively spliced isoforms of BRCA1 (Nelson and Holt, 2010,PMID: 20681793)orhypophosphorylated form of BRCA1 that migrates faster (Nelson et al 1999, PMID: 9988281)(Feng et al 2004, PMID:15087457). BRCA1 doublet on PAGE has also been reported by several other authors (Nelson et al 1999, PMID:9988281)(Feng et al2004, PMID:15087457)(Coene et al, 2005,PMID:15591126)(Nelson and Holt, 2010, PMID: 20681793)(Nelson et al 2010, PMID: 20085797).

Reviewer 2 Report

Investigating the links between BRCA1 and ACC (and fatty acid synthesis) is a very interesting area of research. The manuscript was written well and experiments are done well but there were quite a few differences/inconsistencies in results between the two cell lines and the conclusions may not be fully supported by the data.

Major

For example, in Figures 4C/F the authors state “Immunoprecipitation studies showed that IGF-I reduced the association between BRCA1 and p-ACCA (S79) in high glucose as previously reported, but this was completely reversed with normal glucose conditions (Figure 4C). The same phenotype was confirmed in MCF7 cells, showing that IGF-I induces ACCA dephosphorylation by reducing the association between BRCA1 and p-ACCA (S79) but only in high glucose.” But I am not entirely convinced by the WB data that there is a marked difference between normal and high glucose levels in Fig 4C (T47D cells) and in Fig 4F there are very slight reductions in pACC (IGF+high glucose) and BRCA1 (IGF+normal glucose) under the two conditions.

The authors show a nice schematic overview of their conclusions in Fig 6, but have not directly measured fatty acid synthesis. Without these measurements its hard to determine whether their hypothesis is true i.e. that the ACC/FASN pathway is actually altered in these cells under the different conditions (high/normal glucose, +/- IGF-I).

Also, for Fig 1. ACC blots look convincing but not FASN or pAMPK. 48hr blots (supp fig) are better. 

Minor

Line 19-

 “As previously reported, IGF-I ACCA dephosphorylation…” induced dephosphorylation?

Author Response

Reviewer 2

For example, in Figures 4C/F the authors state “Immunoprecipitation studies showed that IGF-I reduced the association between BRCA1 and p-ACCA (S79) in high glucose as previously reported, but this was completely reversed with normal glucose conditions (Figure 4C). The same phenotype was confirmed in MCF7 cells, showing that IGF-I induces ACCA dephosphorylation by reducing the association between BRCA1 and p-ACCA (S79) but only in high glucose.” But I am not entirely convinced by the WB data that there is a marked difference between normal and high glucose levels in Fig 4C (T47D cells) and in Fig 4F there are very slight reductions in pACC (IGF+high glucose) and BRCA1 (IGF+normal glucose) under the two conditions.

We apologize for a mistake in the labelling for IGF-I in high glucose(Figure 4C). We thank the reviewer for spotting this mistake and we have corrected the mistake in the revised paper. Wehave now added densitometry analysis as a new figure (Figure 4G) to show reversal of the phenotype in IGF-I+high glucose compared to IGF-I+low glucose, and have also amended associated figure legend as appropriate (page 7, line 220 & 226-227).

The authors show a nice schematic overview of their conclusions in Fig 6, but have not directly measured fatty acid synthesis. Without these measurements itshard to determine whether their hypothesis is true i.e. that the ACC/FASN pathway is actually altered in these cells under the different conditions (high/normal glucose, +/-IGF-I).

The reviewer’s point is valid. In the revised manuscriptwehave removed a phrase that suggest that endogenous fatty acid synthesis pathway is altered (line 380). We have also added:Line 307However, these results would have to be confirmed by actual measurement of ACCA activity”.

Line 383-385While our conclusions are based on changes in protein abundance and protein-protein interactions, alterations in the fatty acid synthesis pathway would have to be confirmed by actual measurement of enzyme activity or lipids.

Also, for Fig 1. ACC blots look convincing but not FASN or pAMPK. 48hr blots (supp fig) are better.

We thank the reviewer for the comment which improves the quality of the manuscript. We now show more representativeblots for FASNand pAMPK from replicate experiments(Figure 1D).

Line 19-“As previously reported, IGF-I ACCA dephosphorylation...” induced dephosphorylation?

Line 18-We agree with the reviewer and we thank them for the comment. We havenow added the word “induced” in the revised version.

Reviewer 3 Report

The paper of Koobotse et al., entitled ”Hyperglycaemia impairs BRCA1 in restraining IGF-1 in breast cancer cells” is a very interesting, well designed and  well written paper. I think that it can be accepted in the present form for publication in IJMS.

Author Response

Reviewer 3

The paper of Koobotse et al., entitled ”Hyperglycaemia impairs BRCA1 in restraining IGF-1 in breast cancer cells” is a very interesting, well designed and well written paper. I think that it can be accepted in the present form for publication in IJMS

We thank the reviewer for the nice comments.

Round 2

Reviewer 1 Report

In this revised version, the authors do not consider seriously the concerns that 25mM glucose and serum starvation may have independent effects on the outcome of the experiments not targeted to study of glucose homeostasis (and the potential mitochondrial protector role of IGF-1). Without additional experiments, the conclusions on the role of hyperglycemia seems poorly supported. No improvement has been made in locating AMPK (how is ACCA phosphorylated, if not by AMPK?) or p53, an obvious player in glycolysis and lipid metabolism, on figure 6. While the authors now indicate that the conclusions are based on protein abundance and protein-protein interactions, the loss of interaction between BRCA1 and pACCA is actually not represented in the model on Figure 6 (similarly representation in both conditions), which remains oversimplistic. A better angle for the presentation of the current data may be a description that the conclusions from their previous studies (in 25 mM glucose) are not observed when the cells are grown in normal conditions, with a downplay of the conclusions on what high glucose does as it is not enough elucidated. (Hyperglycemia refers to high levels of sugar in the blood. In in vitro studies, it is more appropriate to use “under high glucose conditions’’).

Reviewer 2 Report

The authors have addressed most of my concerns.